# Acetate-producing bacterium *Paenibacillus odorifer* hampers lung cancer growth in lower respiratory tract: an *in vitro* study

Xiang-xiang Chen,[1,2] Dan Qiu,[2] Yuan Wang,[3] Qing Ju,[2] Cheng-lei Zhao,[4] Yong-shun Zhang,[5] Min Wang,[2] Yong Zhang,[2] Jian Zhang[1,2]

**ABSTRACT** Lung cancer accounts for the large majority of cancer incidence and mortality worldwide for decades. The dysbiotic microbiome and its metabolite secretions in the gut have been regarded as the dominant biological factors in oncogenesis, development, and progression, adding probiotic components of which have come to be potential therapeutic regimes. However, there still exists little knowledge about whether probiotic microorganisms in lower airways inhibit lung cancer by lung microenvironment remodulation. In this study, we performed bioinformatics analysis from previous sequencing data and specific microbiome databases to identify the potent protective microbes in lower airways, followed by bacterial cultivation and morphological verifications *in vitro*. We found that *Paenibacillus odorifer* was correlated closely with the anti-tumorous by-product acetic acid in lower respiratory tract. Additionally, the enrichment of this microorganism in the health, rather than in lung neoplasms from public data sets, further confirmed its protective activity in preserving pulmonary homeostasis. Colony cultivation of this strain and targeted metabolite analysis indicated that *Paenibacillus odorifer* proliferation was weakened at 37°C but lasted longer than it did at the optimal temperature. And performing as a candidate origin of acetic acid, this strain was liable to inhibit the growth of lung cancer cells in time- and dose-dependent approaches which was validated by colony formation assays. These results suggested that *Paenibacillus odorifer* functions as a candidate probiotic in lower airways to restrict lung cancer cell growth by releasing protective molecules, indicating a potential preventive microbial strategy.

**IMPORTANCE** Various types of microorganisms in lower respiratory tracts protect local homeostasis against oncogenesis. Although extensive efforts engaged in gut microbiome-mediated pulmonary carcinogenesis, emerging evidence suggested the crucial role of microbial metabolites from respiratory tracts in modulating carcinogenesis-related host inflammation and DNA damage in lung cancer, which was still not fully understood in lower respiratory tract microbes and its metabolite-mediated microecological environment homeostasis in preventing or alleviating lung cancer. In this study, we analyzed the lower respiratory tract microbiome and SCFAs expression among different lung segments from the same participants, further identifying that *Paenibacillus odorifer* was correlated closely with anti-tumorous by-product, acetate acid in lower respiratory tract by multi-omics analysis. And previous experiments showed this strain could inhibit the growth of lung cancer cells *in vitro*. These findings indicated that *Paenibacillus odorifer* in lower respiratory tracts might perform as a candidate probiotic against lung carcinogenesis by releasing protective factor acetate, which further presented a promising diagnostic and interventional approach in clinical settings of lung cancer.

**KEYWORDS** lung cancer, *Paenibacillus odorifer*, acetate acid, tumor growth, *in vitro* study

Address correspondence to Jian Zhang, zjfmmu19700227@163.com, or Yong Zhang, 15829245717@163.com.

Xiang-xiang Chen, Dan Qiu, and Yuan Wang contributed equally to this article. Author order was determined by their contribution to the article.

The authors declare no conflict of interest.

See the funding table on p. 10.

For recent decades, lung cancer has become the leading cause of cancer-related death globally. Due to embedded with a relatively unidirectional opening environment, frequent exchanges with the outside enable lungs a vulnerability to suffer carcinogenic hazards, where microbiomes are given a biological priority. The steady presence of the microbiome is usually taken as a significant microbial component into consideration by maintaining the host homeostasis, compositional, and relative abundance alterations of which have been widely accepted to contribute to oncogenesis (1–3). With the discoveries of tumor-resident intracellular microbiota (4), the dominant role of microbiome deserved additional spotlights in complex relationships with multiple tumor types, besides gut microbiome (5). Owing to the outburst of microbial identification technologies, the microbiome in lower respiratory tract, which is usually regarded as sterile, has been validated to bear complicated connections with almost all respiratory disorders, including lung cancer (6). Unlike remote microbiome-mediated systemic effects via blood circulation (7), the lower respiratory tract microbiome directly participates in the regulation of local microenvironment within a restricted biomass and further gets entangled with epithelial cells theoretically, which facilitates pulmonary oncogenesis in the absence of integral physical barrier against external atmosphere (8). Indeed, exposure to outside environment makes this open system vulnerable to be dysbiotic in the presence of trifling variations (9, 10). However, there still exists limited knowledge about how the lower respiratory tract microbes take part in local tumorous progress, further exploration of which seemed promising in prevention, diagnosis, and even treatment of lung cancer in clinics (11).

Similarly, the interactive approaches between microbes and the host are also worthy of being valued, partially in that uneven spatial distribution of microbial biomass overloads human cells more than 10 nearly times completely, absolute predominance of which is mainly achieved by probiotics to sustain relatively healthy conditions (11). Except for cumulative cytotoxicity aroused by abnormally overloaded pathogenic microorganism biomass, various existing evidences indicate that probiotics, particularly in restricted environments like lower airways of residents, may block carcinogenesis by balancing immune responses, producing bioactivity-positive probiotic enzymes, and most probably, releasing various metabolites that regulate cellular process in a dynamic manner (12). Among these probiotic metabolites, short-chain fatty acids (SCFA), mainly derived from dietary fiber digestion-associated gut microbiome, have been fully evaluated to perform beneficial effects to prevent tumor initiation and development (13), characterized by inducing apoptosis cancer cells in a ligand-receptor or inhibition of histone deacetylases dependent manner (14). Unfortunately, there still exists a vacuum in exploring the potential correlation of SCFA in lower respiratory tract with corresponding microbiome, protective merits of which may overweight gut microbiome in impeding resident, probably presenting a promising precautious and interventive regimes (15).

Of note, microbial diversity matters relatively compared with bio-loadings under restricted circumstances. Identification of dominant probiotics in lower respiratory tract seemed promising to achieve tumorous hazard assessment and potential preventative microbial interventions among NSCLC sufferers in clinical practice (16, 17). In accordance with Zheng, et al's work (18), our previous work also indicates that the compositional diversity differed in lower respiratory tract of lung cancer patients with the healthy ones (19). More importantly, *Paenibacillus odorifer*, a species of Firmicute in the family *Paenibacillaceae*, enriches in healthy lung microbiome and bears a strong correlation with protective factor of SCFAs, acetic acid (18). Although accounting for a small proportion among lower respiratory tract flora in the above research, several studies have explored its beneficial effects in rework applications (20, 21), but failed to explore its underlying role in cancer blockings, which deserved additional attentions.

In this study, we screened out a set of biomarkers for lung cancer through our previous sequencings from lower respiratory tracts within different lung segments of lung cancer sufferers, preliminarily identifying the tumor inhibitory function of *P.

*odorifer* by producing protective metabolite, acetic acid *in vitro* study (19). Specific strain cultivation under temperature discrepancy further indicated a suitable position for *P. odorifer* proliferation at lower airways. The relative long-term presence of *P. odorifer* produced acetic acid in a time-dependent manner, by which this strain may inhibit lung cancer cells growth *in vitro*, presenting a promising precautious and interventive strategy to lung cancer in clinics.

## RESULTS

### *Paenibacillus odorifer* in healthy lungs correlated with acetate acid

It has been widely accepted that the reduced microbial diversity in lower respiratory tract may contribute to oncogenesis by activating several tumor growth-related cascades within cells (22). According to our previous study, we also found that the microbiome differed in health and tumor burden lung segments within the same patients. By comparing the metagenomic sequencing and targeted metabolome of health and tumor burden lung segments within the same patients, coupled with combined analysis of candidate microbes and differentiated acetate acid, we found that *Paenibacillus odorifer*, which was enriched in the healthy lungs, correlated closely with acetate acid by spearman correlation analysis (Fig. 1A). And ellipse heatmap further validated the potential correlation between them both, indicating a probiotic effect to inhibit oncogenesis to some extent (Fig. 1B). Additionally, linear correlation of indicated microbe with acetate acid among all enrolled patients indicated a strong correlation ($r = 0.303$, *$P = 0.02598$, Fig. 1C) compared with other candidate microbes (Fig. S1), further supporting the perspective that *Paenibacillus odorifer* in lower respiratory tract may be regarded as a relevant biological origin of acetate acid.

### *Paenibacillus odorifer* sustains lower airway homeostasis as a protective element

To further explore the probiotic role of *Paenibacillus odorifer* in lower respiratory tract, we screened out the compositional diversity of indicated bacteria in lower airways of systemic diseases on Human Microbiome Bodymap (mBodyMap; www.mbody-map.microbiome). Accordingly, *Paenibacillus odorifer* often performs as a protective element against multiple diseases, including respiratory infection, inflammatory bowel diseases, and other systemic disorders, mainly dependent on the relative abundance and compositional alterations as data set on this website (Fig. 2A). In pulmonary microenvironment, *Paenibacillus odorifer* colonized in healthy lungs primarily, which could be switched in pneumonia, partially explained by specific bacteria dominating conditionally and overshadowing its abundance (Fig. 2B). As for those in neoplasms, a decrease of *Paenibacillus odorifer* seemed to be put under the spotlight to the correlation with tumor initiation despite of loaded in a restricted abundance, which was verified at diversity comparison at genus and species level, respectively (Fig. 2C and D). Taken together, these results indicated that *Paenibacillus odorifer* is possibly taken as a protective microbial markers, sustaining lower respiratory tract homeostasis from oncogenesis to some extent, which may partially depend on local or remote microbiome contributions.

### Body temperature facilitates *Paenibacillus odorifer* durative growth

As the identified bacteria strain mainly oriented from outside environments in a low biomass, the isolation and culture of which seemed impractical in the complicated microbial presence, type strain was taken as a priority to further explore the inhibitory performance of *Paenibacillus odorifer* in tumor blockade *in vitro*. Due to the conflict of optimal growth temperature at 30℃ normally with that in lower airways, we adopted this strain to 37℃, which was in accordance with human body temperature, and performed colony cultivation to explore the growth condition. Surprisingly, we found that *Paenibacillus odorifer* survived under human temperature for 24 h (Fig. 3A), and gram staining showed the formation of oval rod-shaped spore (Fig. 3B). Additionally, we

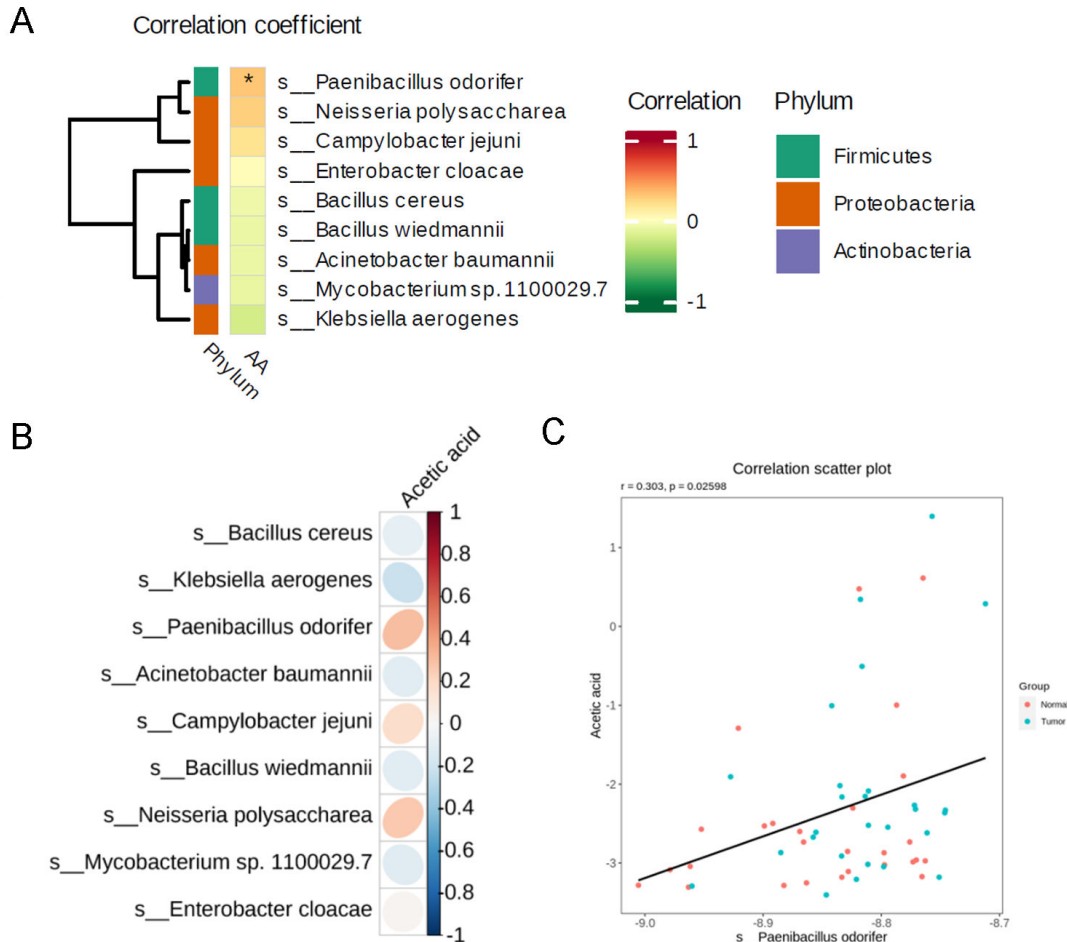

**FIG 1** Dominant role of *Paenibacillus odorifer* and its correlation with acetate acid in lower respiratory tract. (A) Correlation heatmap of candidate microbes in healthy lungs compared with tumor burden lung segments in the same patients. (B) Ellipse heatmap of indicated microbe with acetate acid, respectively. *$P <$ 0.05. (C) Spearman correlation scatter plot of indicated microbe with acetate acid in each samples. $r = 0.303$, *$P = 0.02598$.

also found that human temperature facilitated this strain of *Paenibacillus odorifer* growth at a long term, growth peak of which failed to reach a relative level compared with that under normal conditions (Fig. 3C and D), further explaining that *Paenibacillus odorifer* enriched in healthy lungs with restricted abundance. However, the absorbance of indicated strain at 600 nm showed that those under 37°C overweight those in optimal temperature (Fig. 3E), which could be explained that 37°C flourished type strain but failed to keep them alive. These results illustrated that lower respiratory tract provided a relatively proper temperature for the long-lasting presence of *Paenibacillus odorifer* in a low biomass.

### *Paenibacillus odorifer* derived acetate acid inhibits lung cancer cell growth

Another, we further explored whether acetate acid produced by *Paenibacillus odorifer* acts as a protective factor against lung cancer *in vitro*. Correspondingly, acetate detection assay was launched to examine its content variations in different cultivated conditions under 37°C. According to the results, we found that the concentration of acetate acid sustained at a limited level compared with that in gut, and asynchronous transformative trajectory to bacterial load under different conditions, further illustrating that acetate may act as the metabolite of *Paenibacillus odorifer* under human environment (Fig. 4A). Meanwhile, the peak volume of acetate in 37°C group was lower than that in 30°C group, and the former suffered long lasting effects as the extension of the cultivation period, which could be attributed to nutrient stock aroused by relatively slow growth. Next, we

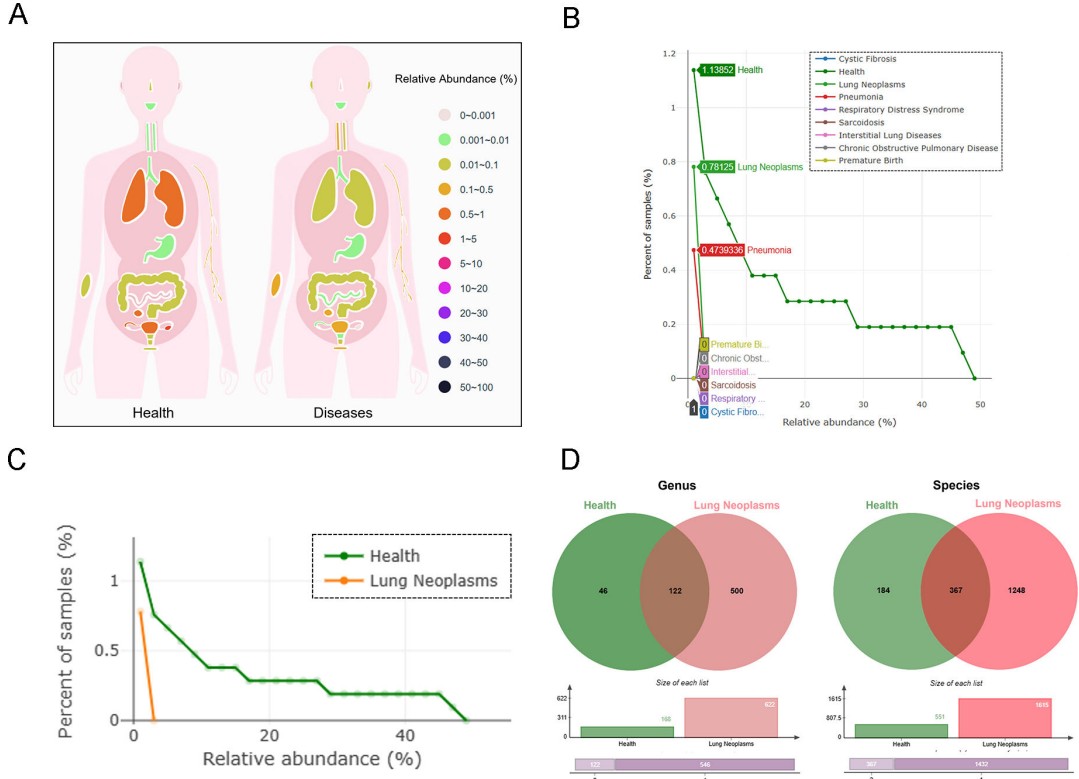

**FIG 2** The probiotic property of *Paenibacillus odorifer* in healthy lungs based on public microbiome database. (A) Relative abundance of *Paenibacillus odorifer* among the health and systemic diseases. (B) Relative abundance of indicated microbe in respiratory disorders according to collective samples. (C) Relative abundance of *Paenibacillus odorifer* among the health and lung neoplasms. (D) Venn plots of differentiated candidates within lung neoplasms and health groups at genus and species level, respectively.

added additional acetate into Beas2B cell and lung cancer cell lines, including PC-9, A549, and H1975, at similar concentration in lower respiratory tract among the enrolled clinical patients, finding that acetate at appropriate concentration restricted lung cancer cell growth under 37°C for 48 h, which could not be reached in 24 h and those in bronchial epithelial cells (Fig. 4B). These results was in line with the abundance variations under the same cultivated conditions, which further indicated that *Paenibacillus odorifer* mediated tumor blockade might be achieved by release of acetate in a dose and time-dependent manner.

## DISCUSSION

Probiotics, previously defined as a combination of live beneficial bacteria and/or yeasts that naturally live in human bodies, accounted for a large majority of integrity microbes in the host, which was extended to guide clinicians and consumers in determining diverse products (23, 24). In light of co-existence with bad bacteria, probiotics guard against those detrimental microorganisms when they overloaded or altered compositionally under infections and other inflammatory disorders (25). Up to now, various studies indicated that probiotics distributed disproportionately (26), mainly enriched in digestive tracts and other sites with close contact to open areas, leading to various contributions to uncover corresponding microbiome with multiple disorders (27). Taking gut microbiome as examples, probiotics in gut flourish at primary sites, which contain the majority of microbes in humans completely, including *Lactobacillus* spp., *Bifidobacterium* spp., *Bacillus* spp., and so forth, and release beneficial metabolites or factors into blood fluid, sequentially transported into remote organs to fight off harmful bacteria or reconstruct inflammatory balance via immunomodulation (28). As it should be that an

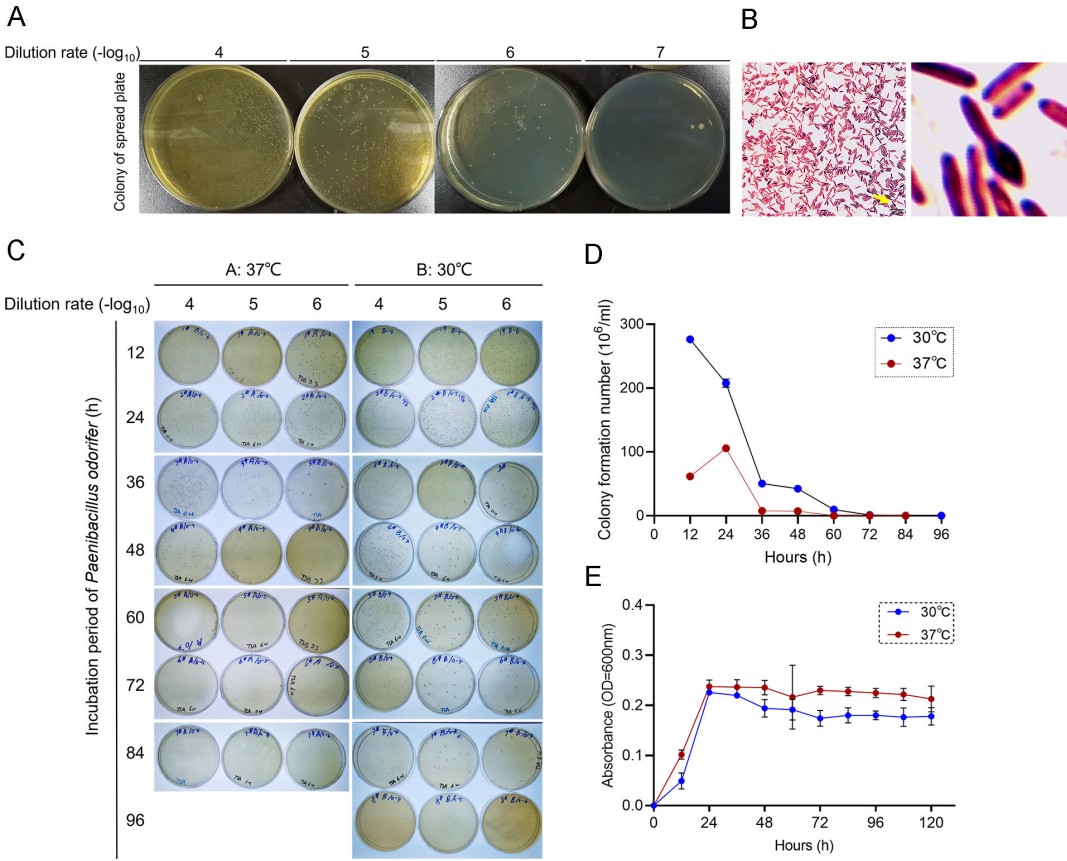

**FIG 3** The human body temperature is suitable for the growth of *Paenibacillus odorifer*. (A) Colony morphology after gradient dilution coating. All plates were phetted in the same condition. (B) Bright-field microscopy of *Paenibacillus odorifer*, showing a rod-shaped and Gram positive, negative, or variable, with peripheral flagella movement (left). Yellow arrow, zoomed oval-shaped spores in the enlarged cyst on the right. Scale bar, 50 µm. (C) Colonies of *Paenibacillus odorifer* at different temperatures with indicated time points according to gradient dilution, respectively. (D) Line chart of statistical colonies at different conditions. Each sample was counted for three times by different operators. (E) Line chart of absorbance of OD at 600 nm within each group.

equilibrium compositional state optimizes probiotic effects in complicated microecological environments, deficiency or redundancy of which may reverse defensive functions to some extent. With respect to low biomass-loaded sites, probiotics appear to play a more substantial role in the management of the local microbiota. Except for indirect modulations driven by lung-gut axis from gut microbiome, probiotics in lower respiratory tract seemingly suffer additional fierce struggles with dynamic microbial alterations at each gas exchanges, compared with those in digestive tracts (3). As for the initiation of lung cancers, microbiome in lower airways (29, 30) prioritizes to impede epithelial cells from pathogenic microorganisms-mediated malignant lesions, however, which failed to reach a consensus. In this study, we found *P. odorifer* categorized into *Paenibacillaceae* family, enriched in healthy lungs and bears strong correlation with probiotic metabolites, further indicating a likelihood in blocking oncogenesis or development of lung cancer despite of consistent efforts into animal husbandry and milk production (20, 21). Although we failed to construct animal models and clinical studies for *in vivo* validation, it still deserves attention on this strain, coupled with other potential lower respiratory probiotics, to uncover its anti-tumor effects as a candidate probiotics in detail.

It is also worth noting that the presence of physical barriers, such as envelopes outside the lesions at the early stages, isolates malignant cells from microbes and the corresponding mucus layer, which process can be transformed as the tumor cells break through the basement membrane into intratracheal spaces (31). During this process, probiotics functions from metabolite- or enzyme-dependent indirect contact to microbe-mediated direct interaction more tendentiously, followed by the reduction of

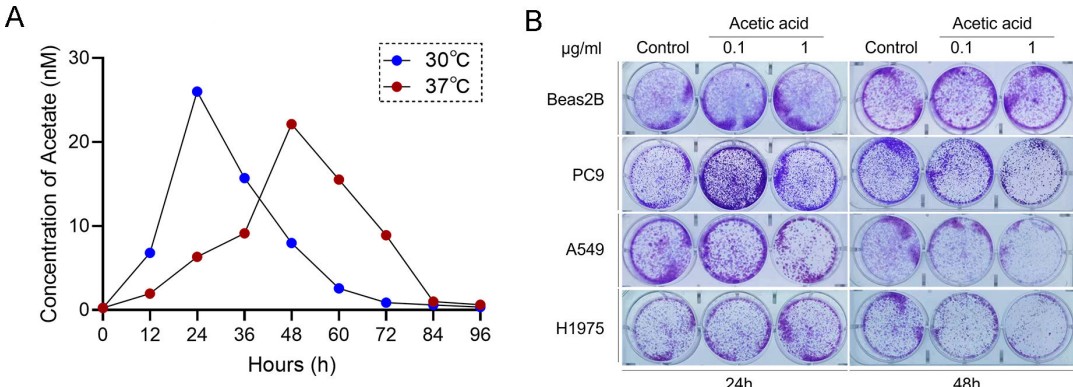

**FIG 4** Bacteria-derived acetate acid inhibits growth of lung cancer cell lines *in vitro*. (A) Acetate acid detection of supernatant within indicated bacterial cultivation. (B) Colony formation of lung cancer cells with the presence of indicated dose of acetate acid for 24 h and 48 h, respectively.

probiotics to some extent (32). Certainly, it still is confusing that whether this abnormal microbial distribution performs as the reason or just the dominant outcomes of oncogenesis or tumor development instead. But the clear one is that these protective products from probiotics function as signals bridging the interaction between microbes and the host, particularly at appropriate concentrations under specific circumstances, where slight fluctuation probably contributed to malignant outcomes (32). Although few studies on probiotics embedded in lower respiratory tracts, several microbial sequencings between the health and malignancy sufferers indicated the diversity and compositional discrepancy of microbiome and relevant metabolome in lower respiratory tract are associated closely with lung cancer at different stages, which further support the perspective that the presence of probiotics in respiratory tract matter partly by releasing dominant probiotic factors to restrict malignant lesions. Additionally, SCFAs, a mainly discussed intermediate products of fatty acid metabolism generated by non-digestive and fermentable carbohydrates from the gut microbiome, have been validated to regulate the physical environment for microbes, maintain inflammatory balance and cellular barrier integrity, and initiate intracellular signaling pathways as potential ligands to membrane or cytoplasm receptors due to their specific characteristics of volitation and solubility (33, 34). From this perspective, we found that *P. odorifer*, the candidate probiotics in lower respiratory tract against oncogenesis, possessed strong correlation with acetate acids, the vast majority type of SCFAs in respiratory airways. The quantification of acetate acids kept in line with the CFUs of indicated strain *in vitro*, tumor inhibition of which was also verified by additional supplement of exogenous acetate acids to lung cancer cell lines. These results illustrated that acetate acids in lower respiratory tract, especially those oriented from local probiotics, may own properties to block mung cancer initiation and development to some extent, which still need further explorations before clinical practices.

In addition, the origins of SCFAs, especially acetate acid explored in this study, should be given a priority to discuss during the exploration of local microbes-mediated probiotic function in lung cancer (35). Due to the restricted carbon atoms under six, acetate acid, which is top-ranked intermediate metabolites of fatty acids with two carbon atoms, is microbially derived from bacteria and production pathways, including endogeneity (hydrolysis of pyruvate via acetyl-CoA) (36), exogeneity (dietary prototype intake) (37), and microbial-derived (Wood-Ljungdahl pathway from bacteria) (38), which is defined as a protective element or energetic source against various diseases besides tumor. Nevertheless, it is this origin diversity from microbes and the host that fails to accurately identify microbial sources in complicated microecological environments. The co-presence of microbes and host cells in lower respiratory tracts coupled with dynamic crosstalk between them both obscure the identification of probiotics-mediated acetate acid release, overlapped by cellular glucometabolic influence. In addition, relatively low

oxygen or anaerobic environment facilitates to convert small molecular metabolites into acetate in the presence of special microbes, which further complex the crosstalk within lower respiratory tract (39, 40). Accordingly, we verified the anti-tumorous effects of indicated metabolite *in vitro*, which might exclude underlying interferences from the host utmost, further illustrating that additional acetate acid may inhibit the proliferation of lung cancer cells and act as a potent probiotic factor of microbes to block tumor development.

However, several limitations still deserve attention in this study. First, complicated interactions between microbes and the host in lower respiratory tract make relative abundances and compositional diversities a priority to explore the protective roles of probiotics in lung, integrity of which may exceed the contribution from a single bacterial species. In this study, we merely presented a possibility that *P. odorifer* may produce acetate acid to inhibit tumor development in specific environments as a potential probiotics, which need to be verified *in vivo* studies. Second, the contribution degree of acetate acids related anti-tumorous effects deserved additional deliberations, partially in that we failed to detect the whole metabolites in lower airways, and the presence of other dominant probiotic factors was not included in this study, weakening the probiotic effects of candidate strain in practice. In addition, retention on the phenomenon of specific strain-derived acetic acid-mediated anti-tumor effects *in vitro,* rather than detailed mechanism exploration, may cripple the perspectives in this study that it still encloses whether *P. odorifer* act as the primary origin of acetic acid and how the latter performs anti-tumor effects under this circumstances. Finally, the absence of experimental animal models for airway microbiota transplantation that simulate the natural state is another key step that hinders the verification of this study before clinical practices.

In conclusion, carcinogenic contributions to cancers from specific microbes have been extensively explored among numerous cancer types, especially various gastrointestinal cancer with gut microbiome. Due to the open environment and dynamic alterations to the outsides with complicated microbial composition, lower respiratory tract microbiome gradually steps into spotlight in recent years. Protective microbes possess abilities to fight with co-present detrimental elements in microbiome, providing promising diagnostic strategies and anti-tumor interventions in the future. In this study, we preliminarily identified a candidate probiotics in lower respiratory tract, *Paenibacillus odorifer*, and its close correlations with potential protective metabolite, acetate, by analyzing the various sequencing data from our previous work and public database. Our findings indicated that *Paenibacillus odorifer* enriched in healthy lungs, maintaining pulmonary homeostasis and protecting lungs suffering neoplasms. Colony cultivation of type strain and indicated metabolite examination under specific conditions further verified that the proliferation of *Paenibacillus odorifer* might be weakened under 37℃ but lasted in the long term compared with that under optimal temperature at 30℃, which might block lung cancer cell growth in a time-dependent manner. Collectively, these findings indicated that *Paenibacillus odorifer* in lower respiratory tracts might act as a candidate probiotic against lung carcinogenesis by releasing protective factor acetate, which further presented a promising diagnostic and interventional approach in clinical settings of lung cancer, still deserving detailed explorations before clinical applications.

## MATERIALS AND METHODS

### Bacterial strain and culture

*P. odorifer* was purchased from GDMCC (Product Strain Number: 1.1144; Guangdong Microbial Culture Collection Center, Guangzhou, China). This strain was cultured under proper aerobic condition in TSA (Tryptase Soya Agar; Sigma Aldrich, Merck KgaA, Darmstadt, Germany) following the manufacturer's recommendations at indicated time points (21). Briefly, the strain was incubated at the proper temperature (30℃ and 37℃, respectively) for indicated time points as mentioned above and, in turn, centrifuged

at 6,000$g$ for 10 min. After repeated washing, bacterial resuspension was performed the optical density (OD600) detection at $10^7$ CFU/mL (colony-forming units per milliliter) (LogPhase 600, Microbiology Reader, Bio-Tek Instruments Inc., Winooski, VT, USA). CFU was accounted according to standard growth curve based on ImageJ (Version 2.14.0/1.54 f, Windows 32bit; NIH, USA). Gram staining was performed using standard microbiological techniques.

## Cell lines and reagents

Bronchial epithelial cells (Beas2B), and NSCLC cell lines, namely, PC-9, A549, and H1975, were purchased from American Type Culture Collection (Shanghai, China) as frozen vials, all of which have been verified by Short tandem repeat. The NSCLC cell lines mentioned above were cultured in RPMI 1640 (Corning). Beas2B were cultured in DMEM (Invitrogen). 10% FBS (Gibco) and 1% penicillin/streptomycin (Invitrogen) were included in these media, and all cells were cultured in an incubator containing 5% $CO_2$ at 37°C. Phosphate saline buffer (PBS), fetal bovine serum (FBS), penicillin and streptomycin mixture, trypsin-EDTA solution, and dimethyl sulfoxide (DMSO) were purchased from Sigma Aldrich (St. Louis, USA). The cell culture media, Gibco RPMI Media 1640 and DMEM were purchased from Thermo Fisher Scientific (NO. 12633012; NO. 12491015, China). All the reagents corresponded to the analytical standard purity and were applied according to the manufacturers' recommendations.

## Acetate quantitative examination

Acetate quantitative examinations were performed by Acetate Colorimetric Assay Kit (No. MAK086; Sigma-Aldrich, St. Louis, USA) according to the manufacturer presented protocol. Briefly, supernatants collected from bacterial cultivation at different conditions suffered 0.22 μm filter membrane to eliminate bacterial interference. Then, samples were deproteinized before additions into a final volume of 50 μL with a reaction system. A total 50 μL of the appropriate reaction mixture was added into each of the wells using a horizontal shaker and, in turn, incubated the reaction for 40 min at room temperature against light during the incubation. Finally, the absorbance at 450 nm (A450) was measured and calculated according to the standardized curve. Each sample was repeated three times.

## Colony formation assay

Cells were seeded into 6-well plates at $1 \times 10^3$ cells per well and incubated at 37°C with 5% $CO_2$ for 4 days. Acetate acid was added into each plate with indicated concentration for different periods (from 0 h to 96 h at 12 h intervals). Then, cells were washed twice with PBS, fixed with 4% paraformaldehyde for 15 min, and stained with crystal violet for 10 min. The number of colonies over 50 cells was digitally counted by a light microscope via ImageJ (Version 2.14.0/1.54f, Windows 32bit; NIH, USA).

## Statistical analysis

The selection of candidate microbes and metabolites was achieved by our previous work (19). Indicated correlation analysis of candidate microbes and metabolites was performed using R packages by Spearman analysis (19). Relevant analytical methods were fully discussed in previous work. The data were processed as means ± standard deviation (SD) by GraphPad Prism (Version 9.0.0 for Windows, GraphPad Software, San Diego, CA, USA). The differences among the groups were compared by performing one-way ANOVA analysis, statistically significant differences of which were labeled with indicated symbols, respectively. Linear regression analyses were performed between different microbial species and SCFAs as previous sequencings and achieved by GraphPad Prism. *$P < 0.05$. $P > 0.05$ equals to NS, no significance.

## ACKNOWLEDGMENTS

We thank the lung cancer patients who participated in this research previously and all the clinical staff that assisted with preliminary processing of this work. This study was funded by grants from the Clinical Booster Project of the Air Force Military Medical University (#2021LC2115) to Jian Zhang.

All authors have read and agreed to the published version of the manuscript. In detail, X.C.: Methodology (equal) writing—original draft preparation (lead). D.Q.: Methodology (lead). Y.W.: Methodology (supporting); investigations (lead). Q.J.: Resources (lead); validation (lead). C.Z.: Validation (lead). Y.Z.: Data curation (lead). M.W.: Software (lead). Y.Z.: Conceptualization (equal); supervision (equal); writing—review and editing (equal). J.Z.: Conceptualization (lead); supervision (lead); writing—review and editing (lead). According to their actual contributions, X.C., D.Q., and Y.W. contributed equally to this work, and were listed as co-first authors in this study. All authors have approved this authorship orders.

## AUTHOR AFFILIATIONS

[1]Department of Pulmonary Medicine, Affiliated Hospital of Northwest University, Xi'an Peoples' Hospital, Xi'an, China

[2]Department of Pulmonary and Critical Care of Medicine, The First Affiliated Hospital of Fourth Military Medical University, Xi'an, China

[3]Department of Microbiology, School of Basic Medicine of Fourth Military Medical University, Xi'an, China

[4]Department of Dermatology, The First Affiliated Hospital of Third Military Medical University, Chongqing, China

[5]School of Basic Medicine, Fourth Military Medical University, Xi'an, China

## AUTHOR ORCIDs

Yong Zhang http://orcid.org/0000-0002-6072-0697

Jian Zhang http://orcid.org/0009-0009-4322-8797

## FUNDING

| Funder | Grant(s) | Author(s) |
| --- | --- | --- |
| Fourth Military Medical University (FMMU) | 2021LC2115 | Jian Zhang |

## AUTHOR CONTRIBUTIONS

Xiang-xiang Chen, Methodology, Writing – original draft | Dan Qiu, Methodology | Yuan Wang, Investigation, Methodology | Qing Ju, Software, Validation | Cheng-lei Zhao, Validation | Yong-shun Zhang, Data curation | Min Wang, Software | Yong Zhang, Conceptualization, Supervision, Writing – review and editing | Jian Zhang, Conceptualization, Supervision, Writing – review and editing

## DATA AVAILABILITY

All data generated or analyzed in this study were oriented from a standardized clinical process and included partially in a previous published article (19). Sequence data that support the findings of this study have been deposited in the NCBI SRA with the primary BioProject accession code PRJNA991321.

## ETHICS APPROVAL

The research presented here was performed in accordance with the Declaration of Helsinki and was approved by the Ethics Committee of the First Affiliated Hospital of the Air Force Medical University (#XJYY-LL-FJ-002). The patients included in this research

signed informed consent based on the voluntary principle before sample collection performance as described previously (19).

## ADDITIONAL FILES

The following material is available online.

### Supplemental Material

**Fig. S1 (Spectrum00719-24-s0001.tif).** Spearman correlation of candidate microbes and acetate acid.

**Supplemental material (Spectrum00719-24-s0002.docx).** Supplementary figure legend.

### Open Peer Review

**PEER REVIEW HISTORY (review-history.pdf).** An accounting of the reviewer comments and feedback.

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
