## [Reviewer comments · Microbiology Spectrum]

Microbiology Spectrum

An acetate-producing bacterium *Paenibacillus odorifer* hampers lung cancer growth in lower respiratory tract: an in vitro study

Xiang-xiang Chen, Dan Qiu, Yuan Wang, Qing Ju, Cheng-lei Zhao, Yong-shun Zhang, Min Wang, Yong Zhang, and Jian Zhang

Corresponding Author(s): Jian Zhang, Xi'an People's Hospital

Review Timeline:

Submission Date:	March 19, 2024
Editorial Decision:	June 18, 2024
Revision Received:	August 1, 2024
Editorial Decision:	August 12, 2024
Revision Received:	August 12, 2024
Accepted:	August 20, 2024

Editor: Dhammika Navarathna

Reviewer(s): The reviewers have opted to remain anonymous.

Transaction Report:

DOI: <https://doi.org/10.1128/spectrum.00719-24>

Re: Spectrum00719-24 (An acetate-producing bacterium *Paenibacillus odorifer* hampers lung cancer growth in lower respiratory tract: an in vitro study)

Dear Prof. Jian Zhang:

Thank you for the privilege of reviewing your work. Below you will find my comments, instructions from the Spectrum editorial office, and the reviewer comments.

Revision Guidelines

Sincerely,
Dhammika Navarathna
Editor
Microbiology Spectrum

Reviewer #1 (Public repository details (Required)):

The authors classified the study as an analysis of available datasets and referred to previous work done in text. However, they neither cited a reference nor provided links to repositories.

Reviewer #1 (Comments for the Author):

The manuscript does not show in-depth or rigorous scientific scrutiny (material and methods) to justify the results or conclusions the authors arrived at. The M & M are not clear for reproducibility. The authors should clarify the weaknesses in their study design. The manuscript requires thorough English language editing and proofreading.

**An acetate-producing bacterium *Paenibacillus odorifer* hampers lung**
**cancer growth in lower respiratory tract: an in vitro study**
**Xiang-xiang Chen^{a,b,#}, Dan Qiu^{b,#}, Yuan Wang^{c,#}, Qing Ju^b, Cheng-lei Zhao^d, Yong-**
**shun Zhang^e, Min Wang^b, Yong Zhang^b, Jian Zhang^{a,b,*}**

5 ^a Department of Pulmonary Medicine, Affiliated Hospital of Northwest University, Xi'an Peoples' Hospital,
Xi'an, China

7 ^b Department of Pulmonary and Critical Care of Medicine, The First Affiliated Hospital of Fourth Military
Medical University, Xi'an, China

9 ^c Department of Microbiology, School of Basic Medicine of Fourth Military Medical University, Xi'an,
China

11 ^d Department of Dermatology, The First Affiliated Hospital of Third Military Medical University, Chongqing,
China

13 ^e School of Basic Medicine, Fourth Military Medical University, Xi'an, China

**Correspondence**
Department of Pulmonary Medicine, Affiliated Hospital of Northwest University, Xi'an Peoples' Hospital,
Xi'an, China; Department of Pulmonary and Critical Care of Medicine, The First Affiliated Hospital of
Fourth Military Medical University, Xi'an, China.
E-mail: zjmmu19700227@163.com (J. Zhang) and 15829245717@163.com (Y. Zhang)
[#]These authors contributed equally to this study.
**ABSTRACT** Lung cancer accounts for the large majority of cancer incidence and
mortality worldwide for decades. The dysbiotic microbiome and its metabolite secretions
in gut have been regarded as the dominant biological factors in oncogenesis,
development and progression, adding probiotic components of which have come to be
potential therapeutic regimes. However, there still exists little knowledge about whether
probiotic microorganisms in lower airways take effects in preventing lung cancer by lung
microenvironment remodulation. In this study, we performed bioinformatics analysis from
previous sequencing data and microbiome database between the health and lung cancer
cohorts to identify the potent protective microbes in lower airways, followed by bacterial
cultivation and morphological verifications in vitro. We found that *Paenibacillus odorifer*
was correlated closely with anti-tumorous by-product acetate acid in lower respiratory
tract. Additionally, enrichment of this microorganism in health lungs, rather than in lung
neoplasms from public datasets, further confirmed its protective activity in preserving
pulmonary homeostasis. Colony cultivation of this strain and targeted metabolite analysis
indicated that *Paenibacillus odorifer* proliferation was weakened at 37°C, but lasted
longer than it did at the optimal temperature. And as a significant source of acetic acid,
this strain was validated to inhibit the growth of lung cancer cells in a time-dependent way
by colony formation assays. These results suggested that *Paenibacillus odorifer* functions
as a candidate probiotics in lower airways to restrict lung cancer cell growth by releasing
protective elements, indicating a potential preventive microbial strategy.
**IMPORTANCE** Various types of microorganisms in lower respiratory tract promoted
sustaining local homeostasis against oncogenesis. Although extensive efforts engaged in
gut microbiome-host dysbiosis, emerging evidence suggested the crucial role of local
microbial metabolites in modulating host inflammation and DNA damage in lung cancer.
In contrast, however, there was little studies focusing on lower respiratory tract microbes
and its metabolites mediated microecological environment homeostasis in preventing or
alleviating lung cancer. In this study, we analyzed the lower respiratory tract microbiome
and SCFAs expression among different lung segments from the same participant, further
identifying that *Paenibacillus odorifer* was correlated closely with anti-tumorous by-
product, acetate acid in lower respiratory tract by multi-omics analysis. And a set of in
vitro studies showed its underlying protective effects on inhibiting lung cancer cells
growth. These findings presented a promising diagnostic and interventional approach
based on *Paenibacillus odorifer* and acetate detection within lower respiratory tract in
clinical settings, which still deserves detailed explorations under adequate examinations.
**KEYWORDS** Lung cancer, *Paenibacillus odorifer*, acetate acid, tumor growth, in vitro
study
For recent decades, lung cancer has become the most diagnosed and primary leading
cause of cancer-related death globally. Due to embedded with a relative unidirectional
opening environment, frequent exchanges with the outside enables lungs a vulnerability to
suffer carcinogenic hazards, where microbiome are given a biological priority. The steady
presence of microbiome is usually taken as a significant microbial component into
consideration by maintaining the host homeostasis, compositional and relative abundance
alterations of which has been widely accepted to contribute to oncogenesis(1-3). With the
discoveries of tumor-resident intracellular microbiota(4), the dominant role of microbiome
deserved additional spotlights in complex relationships with multiple tumor types, besides gut
microbiome(5). Owing to the outburst of microbial identification technologies, the microbiome
in lower respiratory tract, where is usually regarded as sterile, have been validated to bear
complicated connections with almost all respiratory disorders, including lung cancer(6). Unlike
remote microbiome mediated systemic effects via blood circulation(7), lower respiratory tract
microbiome, constructing specific respiratory microecology in a restricted biomass loaded,
directly participates in the regulation of local microenvironment and further gets entangled
with epithelial cells theoretically, which facilitates local oncogenesis with the lack of complete
physical barrier against external atmosphere(8). Indeed, exposure to outside environment
makes this open system vulnerable to be dysbiotic in the presence of trifling variations(9, 10).
However, there still exist limited knowledge about how the lower respiratory tract microbes
take part in local tumorous progresses, further exploration of which hold enormous prospects
of prevention, diagnosis, and treatment in lung cancer clinically(11).

[revised manuscript text omitted]

promising diagnostic and interventional approach based on *Paenibacillus odorifer* and
acetate detection within lower respiratory tract in clinical settings, which still deserves
detailed explorations before clinical applications.
MATERIALS AND METHODS
**Ethics approval and consent to participants.** This research presented here has been performed in
accordance with the Declaration of Helsinki and was approved by the Ethics Committee of the First
Affiliated Hospital of the Air Force Medical University (#XJYY-LL-FJ-002) The patients included in this
research have signed informed consent based on the voluntary principle before sample collection
performance as described previously.
**Bacteria strain and culture.** *P. odorifer* was purchased from GDMCC (Product Strain Number:
1.1144; Guangdong Microbial Culture Collection Center, Guangzhou, China). This strain was cultured
under proper aerobic condition in TSA (Tryptase Soya Agar; Sigma Aldrich, Merck KgaA, Darmstadt,
Germany) following the manufacturers' recommendations at indicated time points. Briefly, the strain was
incubated at proper temperature for indicated time points, and in turn centrifugated at 6000g for 10 min.
After washed with PBS, bacterial resuspension in the PBS was adjusted the optical density (OD600)
detection to correspond to 10⁷CFU/mL (colony forming units per milliliter) (LogPhase 600, Microbiology
Reader, Bio-Tek Instruments Inc., Winooski, VT, USA). CFU was accounted according to standard
growth curve based on ImageJ (Version 2.14.0/1.54f, Windows 32bit; NIH, USA). Gram staining was
performed using standard microbiological techniques.
**Cell lines and reagents.** Bronchial epithelial cells (Beas2B), and NSCLC cell lines, namely PC-9,
A549, and H1975 were purchased from American Type Culture Collection (Shanghai, China) as frozen
vials, all of which have been verified by Short tandem repeat. The NSCLC cell lines mentioned above
were cultured in RPMI 1640 (Corning). Beas2B were cultured in DMEM (Invitrogen). 10% FBS (Gibco)
and 1% penicillin/streptomycin (Invitrogen) were included in these media, and all cells were cultured in
an incubator containing with 5% CO₂ at 37°C. Phosphate saline buffer (PBS), fetal bovine serum (FBS),
penicillin and streptomycin mixture, trypsin-EDTA solution, dimethyl sulfoxide (DMSO), were purchased
from Sigma Aldrich (St. Louis, USA). The cell culture media, Gibco RPMI Media 1640 and DMEM were
purchased from Thermo Fisher Scientific (NO.12633012; NO.12491015, China). All the reagents
corresponded to the analytical standard purity and were applied according to the manufacturers'
recommendations.
**Acetate quantitative examination.** Acetate quantitative examinations were performed by Acetate
Colorimetric Assay Kit (No. MAK086; Sigma-Aldrich, St. Louis, USA) according to manufacturer
presented protocol. Briefly, supernatants collected from bacterial cultivation at different conditions
suffered 0.22µm filter membrane to eliminate bacterial interference. Then, samples were deproteinized
before additions into a final volume of 50µl with reaction system. Total 50µl of the appropriate reaction
mixture were added into each of the wells using a horizontal shaker, and in turn incubated the reaction
for 40min at room temperature against light during the incubation. Finally, the absorbance at 450 nm
(A450) was measured and calculated according to the standardized curve. Each samples were repeated
3 times.
**Colony formation assay.** Cells were seeded into 6-well plates at 1×10^3 cells per well and incubated
at 37°C with 5% CO₂ for 4 days. And acetate acid was added into each plate with indicated
concentration for different periods. Then cells were washed twice with PBS, fixed with 4%
paraformaldehyde for 15 min, and stained with crystal violet for 10 min. The number of colonies that
contained more than 50 cells was digitally counted under a light microscope by ImageJ (Version
2.14.0/1.54f, Windows 32bit; NIH, USA).
**Statistical Analysis.** The selection of candidate microbes and metabolites was achieved by our
previous work. Indicated correlation analysis of candidate microbes and metabolites were performed
using R packages by spearman analysis(19). Relevant analytical methods were fully discussed in
previous work. The data were processed as means ± standard deviation (SD) by GraphPad Prism
(Version 9.0.0 for Windows, GraphPad Software, San Diego, CA, USA). The differences among the
groups were compared by performing one-way ANOVA analysis, statistically significant differences of
which were labeled with indicated symbols, respectively. Linear regression analyses were performed
between different microbial species and SCFAs as previous sequencings, and achieved by GraphPad
Prism. * $P < 0.05$. $P > 0.05$ equals to NS, no significance.
**Data Availability.** All data generated or analyzed in this study were oriented from a standardized
clinical process and included partially in previous published article. Sequence data that support the
findings of this study have been deposited in the NCBI SRA with the primary Bio-project accession code
PRJNA991321 on the following link: <https://www.ncbi.nlm.nih.gov/sra/PRJNA991321>.
**ACKNOWLEDGEMENTS**
We thank the lung cancer patients who participated in this research previously and all the clinical staff
that assisted with preliminary processing of this work. This study was funded by grants from the Clinical
Booster Project of the Air Force Military Medical University (#2021LC2115) to Jian Zhang.
All authors have approved the final version of the manuscript and declared no competing interests.
All authors have read and agreed to the published version of the manuscript. In detail, Xiangxiang
Chen: Methodology (equal) writing—original draft preparation (lead). Dan Qiu: Methodology (lead). Yuan
Wang: Methodology (supporting); investigations (lead). Qing Ju: Resources (lead); validation (lead).
Yang Liu: Validation (lead). Jiaqi Liu: Visualization (lead). Yongshun Zhang: Data curation (lead). Yong
Zhang: Conceptualization (equal); supervision (equal); writing—review and editing (equal). Jian Zhang:
Conceptualization (lead); supervision (lead); writing—review and editing (lead).
REFERENCES
- 1. Sepich-Poore GD, Zitvogel L, Straussman R, Hasty J, Wargo JA, Knight R. 2021. The microbiome
and human cancer. *Science* 371.
- 2. Cullin N, Azevedo Antunes C, Straussman R, Stein-Thoeringer CK, Elinav E. 2021. Microbiome and
cancer. *Cancer Cell* 39:1317-1341.
- 3. Diao Z, Han D, Zhang R, Li J. 2022. Metagenomics next-generation sequencing tests take the stage
in the diagnosis of lower respiratory tract infections. *J Adv Res* 38:201-212.
- 4. Fu A, Yao B, Dong T, Chen Y, Yao J, Liu Y, Li H, Bai H, Liu X, Zhang Y, Wang C, Guo Y, Li N, Cai S.
2022. Tumor-resident intracellular microbiota promotes metastatic colonization in breast cancer. *Cell*
185:1356-1372.e26.
- 5. Chen F, Dai X, Zhou CC, Li KX, Zhang YJ, Lou XY, Zhu YM, Sun YL, Peng BX, Cui W. 2022.
Integrated analysis of the faecal metagenome and serum metabolome reveals the role of gut
microbiome-associated metabolites in the detection of colorectal cancer and adenoma. *Gut* 71:1315-
1325.
- 6. Zhao L, Luo JL, Ali MK, Spiekerkoetter E, Nicolls MR. 2023. The Human Respiratory Microbiome:
Current Understandings and Future Directions. *Am J Respir Cell Mol Biol* 68:245-255.
- 7. Routy B, Le Chatelier E, Derosa L, Duong CPM, Alou MT, Daillère R, Fluckiger A, Messaoudene M,
Rauber C, Roberti MP, Fidelle M, Flament C, Poirier-Colame V, Opolon P, Klein C, Iribarren K,
Mondragón L, Jacquelot N, Qu B, Ferrere G, Clémenson C, Mezquita L, Masip JR, Naltet C,
Brosseau S, Kaderbhai C, Richard C, Rizvi H, Levenez F, Galleron N, Quinquis B, Pons N, Ryffel B,
Minard-Colin V, Gonin P, Soria JC, Deutsch E, Llorca Y, Ghiringhelli F, Zalzman G, Goldwasser F,
Escudier B, Hellmann MD, Eggermont A, Raouf D, Albiger L, Kroemer G, Zitvogel L. 2018.
[revised manuscript text omitted]

508 P values are listed on the top of each plots, respectively.

Dear Editor and Reviewers,

Thank you very much for your time involved in reviewing this original manuscript, **An acetate-producing bacterium *Paenibacillus odorifer* hampers lung cancer growth in lower respiratory tract: an in vitro study (NO. *Spectrum00719-24*)**, and your very encouraging comments on the merits. We also appreciate your clear and detailed feedbacks and hope that the following explanations would fully address all of your concerns. On behalf of all authors of this work, I would like to express my gratitude once again for your kind and constructive suggestions and advises. I stand for all authors to appreciate your concentrations on our work with great pleasure.

As mentioned below, we respond to all reviewers' comments with a point-to-point list of revisions to address your primary concerns. Several improper descriptive errors or opinions, we have launched pertinent revisions. And interpretive questions have been illustrated for details to our best despite of restricted knowledge and scope. We hope these revisions will meet the criterion from journal editor and reviewers.

Each revision was highlighted among Response Letter and marked in the corresponding sites within revised manuscript for clear modified traces. We also welcome suggestive discussion with reviewers on their interests to this manuscript for optimizing our study.

If any questions to our revised manuscript or corresponding fields, please contact us any time. We will provide timely and accurate feedbacks to your concerns. Thanks again for your attentions to our study, and we are looking forward to your reply.

Sincerely yours,

Prof. Jian Zhang,

Lead Corresponding Author

Department of Pulmonary Medicine, Affiliated Hospital of Northwest University, Xi'an Peoples' Hospital, Xi'an, Shaanxi Province, China

Tel: +86-029-84771135

E-mail: zjfmumu19700227@163.com

Reviewer #1:

1. The authors classified the study as an analysis of available datasets and referred to previous work done in text. However, they neither cited a reference nor provided links to repositories. Public repository details (Required)

Response: Thank you for your kind suggestions to our manuscript according to provide public repository details. We also apologized for our negligence to cite referred previous work and links to repositories in advance version. Accordingly, we have added the referred link to our available repositories in the corresponding site in *Marked-up Manuscript*, which has been already deposited in the NCBI Short Read Archive with the primary Bio-project accession code *PRJNA991321* (<https://www.ncbi.nlm.nih.gov/sra/PRJNA991321>) by partial utilization of previous work (*Zhang Y, Chen X, Zhang J, et al. Alterations of lower respiratory tract microbiome and short-chain fatty acids in different segments in lung cancer: a multiomics analysis. Front Cell Infect Microbiol. 2023 Oct 16;13:1261284. PMID: 37915846*), and cited public datasets website in *Materials and Methods* (<https://mbodymap.microbiome.cloud/>). We did acknowledge that the data in this research have not been published in our previous work. We thanked for your constructive suggestions to our manuscript again, hoping these revisions will meet your criteria for further publication at *Microbiology Spectrum* in next revisions.

2. The manuscript does not show in-depth or rigorous scientific scrutiny (material and methods) to justify the results or conclusions the authors arrived at. The M & M are not clear for reproducibility. The authors should clarify the weaknesses in their study design. The manuscript requires thorough English language editing and proofreading. (Comments for the Author)

Response: We stand for all authors to accept your concerns to rigorous scientific scrutiny in corresponding sections, especially in *Materials and Methods*. To improve accuracy and scientific portraits of this manuscript as Reviewer recommended, we further clarified the weaknesses in this study design by adding several key method details to provide further reproducibility at corresponding section as shown in *Marked-up Manuscript*. Additionally, as for the English language editing, we attached great importance to this issue by comprehensive modifications and careful scrutiny, in order to provide a relative flawless manuscript version after detailed proofreading. Revisions have been marked at corresponding sites in revised manuscript. Thank you for your kind suggestions and patience to our work again, and we hope our revisions to your recommendations will contribute to smooth publication of this manuscript.

Re: Spectrum00719-24R1 (An acetate-producing bacterium *Paenibacillus odorifer* hampers lung cancer growth in lower respiratory tract: an in vitro study)

Dear Prof. Jian Zhang:

Thank you for the privilege of reviewing your work. Below you will find my comments, instructions from the Spectrum editorial office, and the reviewer comments.

Revision Guidelines

Sincerely,
Dharmika Navarathna
Editor
Microbiology Spectrum

Reviewer #1 (Comments for the Author):

Comments for Authors

The right articles are missing in several places

Line 22 add article ... the; line 23 add comma before and; line 27 change to databases

Line 30 add article ... the enrichment; Lines 43 ... change to microbiome-mediated

Line 46 ... change to metabolite-mediated;

Line 51-52 ... rephrase "And a set of experiments showed this strain could inhibit lung cancer cells growth in vitro". Consider 'previous experiments showed this strain could the growth of inhibit lung cancer cells in vitro'.

Lines 59-61 rephrase

Line 61 change to microbiomes; line 68 change where to which

Line 77 change progresses to progress; line 78 rephrase "which hold enormous prospects of prevention, diagnosis, and treatment in lung cancer clinically"

Line 82 relative or relatively; 83; abnormal or abnormally; 84 environment or environments

Line 91 exist or exists; line94: remove of ..."cancer despite of low biomass"

Rephrase line 69-74 for clarity

Page 3 line 04 change among to of.

Under results page 4

Line 16 change from health lungs to healthy lungs; line 18: growth-related

Page 5 ... line 62: relative or relatively; rephrase line 62-64

Line 82: time-dependent (add hyphen); line 88 are overloaded; line 93 wrong verb correct 'Taken'; line 94 human or humans?

Page 7 .. line 48: top-ranked; line 50 change to microbially derived

Page 8 ... line 64 rephrase "However, there still exists several limitations to be not negligent in this study."

Line 74 & 75 : strain-derived and acid-mediated

Under Methods page 9

Line 06 add citation; line correct to manufacturer's; line 22: remove 'with'; line 25: correct 'thermo'; line 32 add A total; remove 'and' line 38

Page 10 ... line 345 change spearman to Spearman

Dear Editor and Reviewers,

Thank you very much for your time involved in reviewing this original manuscript, **An acetate-producing bacterium *Paenibacillus odorifer* hampers lung cancer growth in lower respiratory tract: an in vitro study (NO. *Spectrum00719-24R1*)**, and your very encouraging comments on the merits. We also appreciate your clear and detailed feedbacks and hope that the following revisions would fully address all of your concerns. On behalf of all authors of this work, I would like to express my gratitude once again for your kind and constructive suggestions and advises. I stand for all authors to appreciate your concentrations on our work with great pleasure.

We feel sorry for our negligence to missing or inappropriate writings in previous version. As mentioned below, we respond to all reviewers' comments with a point-to-point list of revisions to address your primary concerns. Several improper descriptive errors or opinions, we have launched pertinent revisions. And interpretive questions have been illustrated for details to our best despite of restricted knowledge and scope. We hope these revisions will meet the criterion from journal editor and reviewers.

If any questions to our revised manuscript or corresponding fields, please contact us any time. We will provide timely and accurate feedbacks to your concerns. Thanks again for your attentions to our study, and we are looking forward to your reply.

Sincerely yours,

Prof. Jian Zhang,

Lead Corresponding Author

Department of Pulmonary Medicine, Affiliated Hospital of Northwest University, Xi'an Peoples' Hospital, Xi'an, Shaanxi Province, China

Tel: +86-029-84771135

E-mail: zjfmumu19700227@163.com

Reviewer #1:

1. The right articles are missing in several places.

Response: Thank you for your careful review and patient guidance to our manuscript, especially in scrutinizing the miswritings in this version. As the Reviewer recommended, we have revised these inappropriate writings in the corresponding sites. Due to the commonality of these mistakes, we failed to present point-to-point revisions, but to revise these errors within a integrated response. Corresponding revisions have been achieved in the marked-up manuscript. We hope these revisions will meet your demand and this manuscript will be accepted soon once upon it conforms to publication requirements after further revisions.

Re: Spectrum00719-24R2 (An acetate-producing bacterium *Paenibacillus odorifer* hampers lung cancer growth in lower respiratory tract: an in vitro study)

Dear Prof. Jian Zhang:

Your manuscript has been accepted, and I am forwarding it to the ASM production staff for publication. Your paper will first be checked to make sure all elements meet the technical requirements. ASM staff will contact you if anything needs to be revised before copyediting and production can begin. Otherwise, you will be notified when your proofs are ready to be viewed.

Sincerely,
Dharmika Navarathna
Editor
Microbiology Spectrum